# Assessing the Seafood Trade Diversion Arising from Economic Sanctions: Evidence from Russia and Western Countries

**DOI:** 10.3390/foods12213934

**Published:** 2023-10-27

**Authors:** Chang Min Kim, Dae Eui Kim, Song Soo Lim

**Affiliations:** Department of Food and Resource Economics, Korea University, Seoul 02841, Republic of Korea; kim_changmin@korea.ac.kr (C.M.K.); es992306@korea.ac.kr (D.E.K.)

**Keywords:** trade sanction, trade deflection, seafood trade, gravity model, Russia

## Abstract

Since 2014, economic sanctions between Russia and Western nations have significantly altered the global seafood trade. The consequent decline in bilateral trade also had spillover effects on the rest of the world (ROW). According to earlier studies, economic sanctions appear to negatively impact bilateral trade and income. However, few studies examine how Russian sanctions affect the world as a whole and estimate their effects on the fisheries industry. This study seeks to close this gap by quantifying the extent to which Russian sanctions have impacted trade in terms of trade deflection, trade destruction, trade depression, and trade creation. To this end, panel data from 185 countries were created for the years from 2005 to 2020. With trade policy variables that account for changes in trade channels, a structural gravity trade model was specified. Based on calculations using the Poisson pseudo-maximum likelihood (PPML) fixed effect model, economic sanctions led to a 119.28% surge in Russia’s seafood imports from the rest of the world (ROW), alongside a 39% decline in imports from Western countries. The extent of trade deflection, which includes the exports of Western nations diverted from Russia to the ROW markets, increased by 5.49%. The results demonstrate that trade between sanctioned states, as well as global trade, is significantly impacted by economic sanctions.

## 1. Introduction

Economic sanctions are one of the primary non-military tactics used by nations to voice dissatisfaction, exert pressure, or demand behaviour change from target nations. They refer to the intentional, government-instigated withdrawal or threat of withdrawal of customary trade or financial relations [1]. Recent exchanges between Russia and Western nations, particularly in relation to seafood, are noteworthy examples of such sanctions.

Increased tensions between Russia and the West resulted from a sequence of events in 2014 surrounding Russia’s annexation of Crimea. At the time, most states did not accept the annexation of Crimea, evidenced, for instance, by the adoption of United Nations (UN) General Assembly resolution 68/262, entitled ‘Territorial Integrity of Ukraine’, on 27 March 2014 [2]. Western nations, led by the EU and the US, responded with a broad range of sanctions aimed at several facets of the Russian economy. Statista (2023) [3] estimates that the EU and the US slapped 2099 and 1155 sanctions on Russia between February 2022 and February 2023, respectively.

However, these measures were mutually leveraged, with Russia retaliating with its own set of counter-sanctions [4]. In this regard, Russian counter-sanctions targeted three areas [5]. First, Russia implemented an embargo on food and agricultural products. Second, Russia boosted its domestic production to reduce its dependency on agricultural imports. Third, it expanded its trading relationship with different trading partners, primarily China and the Eurasian segment of the EU. The imposition of the agri-food import No changes required restriction surprised many because of Russia’s heavy reliance on imports from the West. The embargoed goods, which included milk, dairy products, fruit, vegetables, meat, and fish, constitute more than half of Russia’s agri-food imports and are worth close to USD 10 billion [6].

Over the past two decades, Russia’s trade policy has shifted from one of free trade to one of a more regulated trading system, largely due to an increase in domestic supply and a decrease in reliance on imports. For instance, based on the Food Security Doctrine (FSD) established in 2010, Russia steadily boosted its local food supply while bolstering the central government’s ability to control trade and impose restrictions on imports from other nations [7]. By 2022, the FSD intended Russia to be self-sufficient with regards to food [8]. The 2020 revised version includes a more thorough and effective food security plan as well as a preferred course for the food industry’s future development.

The seafood trade sector is one of the sectors significantly affected by these punitive counter-sanction strategies. Fish and seafood exports from Russia have benefited greatly from the country’s extensive coastline, particularly in the North Pacific [9]. Any sanctions affecting Russian seafood were sure to cause repercussions across global trade networks due to its crucial role in the world market. A key goal of this study is to examine how sanctions imposed by Western nations and counter-sanctions imposed by Russia have altered the pattern of seafood trade.

The overarching objective of this study is to conduct an empirical investigation of how economic sanctions impact the global seafood trade. Therefore, this study explores potential changes in trade pathways between the sanctioned states, Russia, and Western economies, as well as their interactions with the ROW, within the framework of a structural gravity model of trade.

The structural gravity model is particularly suited for trade analysis, as it allows for the comprehensive assessment of both observable and unobservable factors influencing international trade flows. The model is calibrated using panel data, encompassing a broad range of seafood exports between various countries over a 2005–2020 period. This methodological approach enables us to isolate the impact of economic sanctions from other potential confounding variables, thereby offering robust estimates of trade diversion in the seafood market.

## 2. A Sanction-Induced Change in the Seafood Trade

Table 1 shows the market shares of top seafood exporters to Russia.

Prior to the 2014 sanctions, Norway was a large supplier of seafood (mainly Atlantic salmon) to Russia. The reciprocal trade ties in the fishing industry were so strong that Norway supplied a sizeable share of Russia’s purchases of seafood [10]. EU nations, especially the Baltic republics like Estonia and Latvia, also had active trade ties with Russia [11]. 

Following the 2014 sanctions, seafood imports from Norway and the EU were significantly reduced by Russia’s punitive counter-sanctions, creating an opportunity for other nations to fill the resultant void. For instance, China and Chile have become important suppliers. China profited from its closeness and diversified fisheries, and Chile, with its thriving salmon-farming sector, emerged as a supply alternative to the predominately Norwegian suppliers. Additionally, Turkey has significantly improved its market access to Russia, which is considered to be a structural development. The growth of Belarus’s exporting role is an intriguing finding. Despite being a landlocked economy, which makes exporting challenging, this nation has made significant strides in seafood exports, which suggests that Belarus’s function as a middleman between Western nations and Russia has grown more crucial in the process of trade [12,13,14]. 

The EU was a significant market for Russian seafood exports before the 2014 sanctions, with a preference for herring, mackerel, and cod [15]. Given their proximity to Russia and seafood-heavy diets, South Korea and Japan were also key Asian markets for Russian seafood [16]. Due to the weakened trade ties with the EU following the 2014 sanctions, Russia began looking for new markets for its seafood. As a result, China became the main market for Russian seafood. Increased fish and seafood trade was made possible by the Belt and Road Initiative’s extension and the improvement of Sino–Russian bilateral relations [17]. As Russia looked to broaden its market access and lessen its reliance on old trading partners, exports to nations in Africa and the Middle East also saw an increase.

This study develops four hypothetical scenarios, shown in Figure 1, to better understand the intricate web of trade results. In this regard, it is assumed that three nations are participating. Western nations and Russia are non-cooperative powers that impose reciprocal economic restrictions on one another. The rest of the world (ROW)—third parties—constitutes the group of non-participating states that abstains from issuing or supporting the sanctions and preserves impartial trade ties with both Russia and the West.

First, trade destruction happens when sanctions are implemented and bilateral trade decreases. As a result of these sanctions, both Russia and the West cease importing seafood. The amount of trade between the two non-cooperating states consequently decreases. 

Second, trade deflection (diversion) refers to the diversion of trade from a sanctioned to a non-sanctioned pathway. To explain: Russia used to import particular seafoods from Western economies. However, Western nations began shipping the same seafood to the ROW as a result of the sanctions. Thus, due to these restrictive trade policies, trade has been ‘deflected’ from Russia to third parties.

Third, trade creation describes the development of fresh trade relationships or the reinforcement of current ones as a result of the sanctions. Russia imported a relatively tiny amount of seafood from the ROW prior to the restrictions. The sanctions with Western nations forced Russia to step up its trade with the ROW, increasing the amount and diversity of seafood.

Finally, a trade depression happens when a country experiences decreased trade, possibly due to decreased capacity, unpredictability, or other associated issues, not just with the one implementing the sanctions but also with other countries. As a result of decreased export capacity or increased uncertainty brought on by Western trade policy, neutral third parties, which are not subject to sanctions, may decrease their trade volumes with the West.

## 3. Literature Review

The body of literature regarding the effects of sanctions on trade is extensive, diverse, and ever-evolving. Although the impact of sanctions on target economies can be significant, they may still have unintended side consequences that extend across international trade networks and impact both the imposers and targets of these economic measures.

Table 2 presents the summary of some studies on the consequences of economic sanctions. Most studies conclude that economic sanctions hurt economies by negatively affecting their income and trade [18,19,20,21,22]. Due to variations in the coverage, severity, and duration of sanctions, it is nearly impossible to assess their absolute impact. However, the impact of sanctions on the economy seems moderate. Another intriguing conclusion is that, in contrast to sanctions that were actually implemented, threats of sanctions encouraged exports [23].

Table 3 reflects selected studies that investigate how trade policy restricts market access to imposing nations, which results in the diversion of exports to markets in other parts of the world. This deflection can be facilitated using a number of trade policy tools, such as anti-dumping duties and import refusals based on concerns about food safety or unapproved genetically modified (GM) or novel foods.

While this study is consistent with previous research, it differs in two areas, one of which is its emphasis on the trade in only seafood. The majority of two-digit HS 03 products have been impacted by Russian counter-sanctions. Therefore, an empirical result could show how trade impacts and trajectories have changed throughout the whole fishery sector. The other is to employ balanced panel data, which includes worldwide bilateral trade flows and is, thus, likely to yield robust evidence from empirical research.

## 4. Model Specifications and Data

The structural gravity model is defined by Equation (1) [27].
(1)$$X_{ij}=\frac{Y_{i}Y_{j}}{Y^{k}}{\left(\frac{t_{ij}}{\prod_{i}P_{j}}\right)}^{1-\sigma }$$
where $\prod_{i}={\left\{{\sum_{j}\left(\frac{t_{ij}}{P_{j}}\right)}^{1-\sigma }\frac{Y_{j}}{Y^{k}}\right\}}^{\frac{1}{1-\sigma }}$ and $P_{j}={\left\{{\sum_{j}\left(\frac{t_{ij}}{\prod_{i}}\right)}^{1-\sigma }\frac{Y_{i}}{Y^{k}}\right\}}^{\frac{1}{1-\sigma }}$.

$X_{ij}$ is the bilateral trade flows between countries i and j; $Y_{i}$ is the income of country i; $Y_{j}$ is the expenditure in country j; $Y^{k}$ is the total world output in sector k; $t_{ij}$ is a bilateral trade cost; $\pi_{i}$ and $P_{j}$ are multilateral resistance terms for countries i and j, respectively, representing all the obstacles and factors that prevent the country from importing or exporting. Finally, σ is the elasticity of substitution among products from different countries.

The cost of bilateral trade, $t_{ij}$, includes direct demand shifters, like tariff equivalents or bilateral trade protection; non-tariff barriers, like the bilateral distance between the two countries (DIST); and common dummy variables, like contiguity (BOR), language (LNG), colonial history (COL), and free trade agreements (FTAs). The cost of bilateral trade is, thus, specified as:(2)$${t_{ij}}^{1-\sigma }=(1+{\tau_{ij})}^{\gamma_{1}}{{FTA}_{ij}}^{\gamma_{2}}{{DIST}_{ij}}^{\gamma_{3}}e(\gamma_{4}{BOR}_{ij}+\gamma_{5}{LNG}_{ij}+\gamma_{6}{COL}_{ij})$$
where ad valorem most favoured nation (MFN) tariffs are represented by $\tau_{ij}$. Because MFN tariffs are included as a regressor, it is possible to directly estimate the elasticity of substitution from the estimate of $\gamma_{1}=1-\sigma$. 

The following can be demonstrated by replacing Equation (2) for $t_{ij}$ in Equation (1) and then expanding the empirical equation into a panel setting with a multiplicative error term:(3)$$X_{ijt}=\beta_{0}+\beta_{1}ln{GDP}_{it}+\beta_{2}ln{GDP}_{jt}+\beta_{3}ln{DIST}_{ij}+\beta_{4}{BOR}_{ij}+\beta_{5}{LNG}_{ij}+\beta_{6}{COL}_{ij}+\beta_{7}{FTA}_{ij}+\beta_{8}{T\_DES}_{ijt}\;+\beta_{9}{T\_DEF}_{ijt}+\beta_{10}{T\_CRE}_{ijt}+\beta_{11}{T\_DEP}_{ijt}+u_{ijt}$$

Two changes were made to the theoretical specifications in the final version of the econometric gravity Equation (3): (i) the omission of MFN tariffs because seafood is defined as a two-digit HS 03 product with an aggregated nature and (ii) the addition of trade dummy variables in order to capture the effects of economic sanctions on trade [14,25,26]. 

The explanations for each variable in Equation (3) are given in Table 4. Finding the effects of the change in trade directions is of research interest. Russian counter-sanctions would indicate the average impact of trade, with everything else remaining constant. These sanctions are expected to be reflected by the four dummy variables created by the theory. For trade deflection and creation, the expected signs for coefficient estimates are positive. However, for trade destruction and trade depression, the expected signs are negative.

It should be noted that the specific error term in Equation (3), $u_{ijt}$, indicates unobserved transaction costs, which can lead to biased estimations. So, applying the fixed effect (FE) will ensure that unobserved time-invariant elements have no influence on the results.
(4)$$u_{ijt\;}=\gamma_{it}+\eta_{jt}+\mu_{ij}+\varepsilon_{ijt}$$
where $\gamma_{it}$ and $\eta_{jt}$ are the country–time variant FE for countries i and j, respectively; $\mu_{ij}$ is the country–pair specific FE that captures all unobserved, time-invariant transaction costs and heterogeneity between countries i and j; and $\varepsilon_{ijt}$ is the error term. The FE, $\mu_{ij}$, makes sure that any unobserved factors that have an impact on trade between country pair i and j but do not change over time are taken into account [27,28,29]. Fally (2015) [30] notes that the PPML with the FE can produce structurally consistent estimates if the model’s underlying structural assumptions are correct. 

For 185 nations from 2005 to 2020, panel data with HS Code 03 seafoods are established. About 80% of the total 544,640 observations have zero trade values. The UN’s Comtrade dataset is used to build the values of bilateral trade, while the International Monetary Fund (IMF) provides statistics on country-level gross domestic product (GDP). The Centre d’Etudes Prospectives et d’Informations Internationales (CEPII) offers information on non-tariff transaction costs, such as geographical distance, contiguity, common languages, and colonial past.

Between 2014 and 2016, Russia implemented counter-sanctions on a total of 37 nations. Four dummy variables are built using 36 of these sanctioned nations, excluding Liechtenstein, to capture the diverse trade effects.

Table 5 provides a data description summary.

## 5. Estimation Results and Discussion

The estimation results are displayed in Table 6. The robust standard errors (SEs) were estimated to counter heteroskedasticity. While Models (c) and (d) represent the outcome of country–time FE, and country–time and country–pair FE, respectively, Models (a) and (b) are coefficient estimates without FE. 

The majority of the estimated coefficients in Models (a) and (b) accord with the predictions made by the theory. Exporters’ and importers’ GDPs, which measure suppliers’ capacity for production and buyers’ demand for goods and services, respectively, are both positive and statistically significant. According to Model (b), ceteris paribus, a 1% rise in the exporter’s and importer’s GDP results in increases in trade of 0.89% and 0.56%, respectively.

The projected adverse impact on trade is demonstrated by using geographic distance as a proxy for various transaction costs in trade. Common language exhibits a negative coefficient estimate, going against the idea that it should encourage more trade. Interestingly, under the FE estimation in Model (c), common language results in a positive coefficient estimate. The FTA, as a tool for trade facilitation, seems to promote trade between member states. However, FE Model (d) indicates that the FTA variable is neither statistically significant nor positive.

In terms of pseudo R^2^, FE Models (c) and (d) have 25% and 39% more explanatory power, respectively, than Models (a) and (b). The FE models offer more accurate estimation findings by controlling unobserved country-specific trends and time-invariant aspects of bilateral relations. The following calculation can be used to determine the size of the trade impact, ${TI}_{i}$, caused by the sanctions based on their coefficient estimates, $\beta_{i}$.
(5)$${TI}_{i}=\left\{\exp\left(\beta_{i}\right)-1\right\}\times 100\%$$

Figure 2 shows the trade impact calculated from each coefficient estimate per trade pathway using the FE.

According to Model (d), which estimates trade destruction at up to −89% on average, Russian imports of seafood from Western nations have all but vanished. Russia expanded trade with the ROW, comprising non-participating nations, to compensate for its losses with the West. The value of imports from the ROW has increased by 118% as a result of the so-called trade creation. Russia virtually replaced seafood imports from Western nations with imports from non-sanctioned nations by fostering its trade relations with the ROW.

Likewise, Western economies adjusted their trade more in the direction of the ROW. The effects of trade deflection and depression can be used to explain this trade divergence. Because Model (c), under the country–time FE, was not statistically significant, only Model (d) calculated the trade deflection estimate as a 5% increase in exports by Western economies. The 5% rise in exports may appear minor at first glance; however, it can be inferred that this is because more Western exports are going to other Western nations. Finally, Model (c) shows that the trade depression effect causes exports to increase by 51%. It is evident that the economic sanctions have increased the role of non-participating nations as substitute seafood suppliers on a global scale.

Overall, the empirical research offers robust evidence that economic sanctions directly affect the ability of sanctioned states to trade. At the same time, the economic sanctions have a knock-on impact on other nations, making sweeping as well as substantial changes in global trade possible.

## 6. Conclusions

The global seafood trade has seen significant changes and instability since Western nations began imposing economic sanctions on Russia in 2014, which were followed by Russian counter-sanctions. The long-term alliance between Russia and the West, including Norway and the EU, was abruptly broken, and other nations, such as China, Chile, and Turkey, became important seafood suppliers to Russia. However, these structural changes have not only affected trade but also affected domestic sectors. By implementing the 2010 and revised 2020 Food Security Doctrine, Russia has continuously strengthened its food self-sufficiency plans and eventually increased its domestic supply base and export capabilities. In essence, Russia was able to create more economic activity with non-participating nations than it lost to trade destruction.

The defensive response by the West to Russian counter-sanctions has also been rational. They have shifted seafood exports from Russia to unsanctioned countries and their internal markets. It is clear that Western nations were able to overcome the barrier to market access and even attain ‘a bigger pie’ by expanding their trade networks to other economies.

The ‘multilateral resistance term’ serves as a reminder of how the ROW is essential in reducing the risks associated with economic sanctions [27,31,32]. These nations are involved in a variety of trade dynamics, including trade creation, trade reflection, and trade depression.

Results derived from the gravity model emphasize that the economic sanctions have influenced trade routes beyond just the relationship between Russia and Western nations, extending their impact to trades involving the ROW. Following the sanctions, Western nations’ exports to the ROW rose by 5% while their imports from the ROW rose by 51%. When Russian seafood imports from Western nations nearly completely disappeared, Russia increased its trade with the ROW by 118% to make up for its losses with the West.

The clarification of how the international seafood trade is connected to economic sanctions and the quantification of the trade effects of categorized trade channel alterations, including trade destruction, trade deflection, trade depression, and trade creation, are two concrete contributions of this study’s empirical investigation. Nonetheless, more investigation that goes as far as disaggregated seafood, such as Norwegian salmon or herring, would facilitate a better understanding of the effects of economic sanctions. Additionally, it will be helpful to look at specific and varied issues regarding the effects of trade from the perspectives of short- and long-term changes in trade, global market instability, the consequences of a change in trade relations, mismatches in seafood quality and standards, and environmental implications.

## Figures and Tables

**Figure 1 foods-12-03934-f001:**
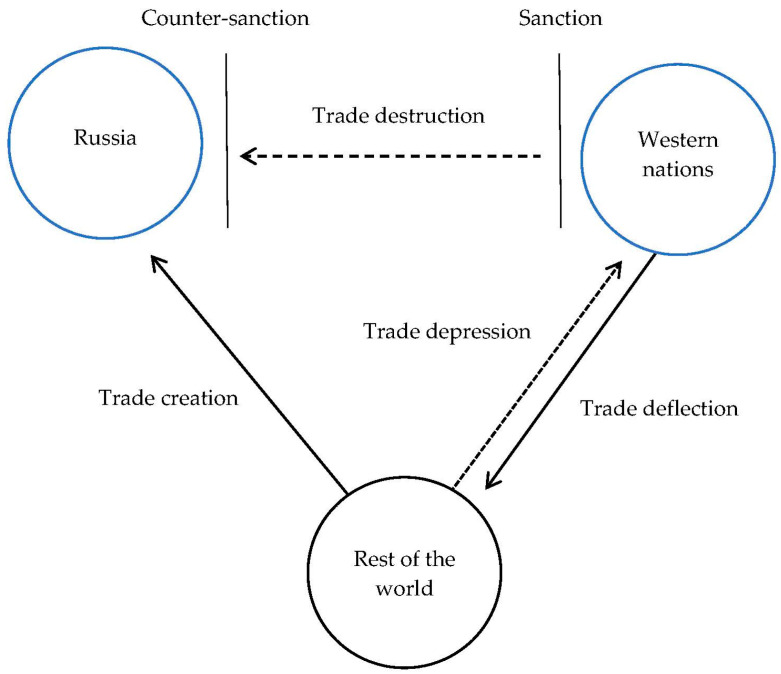
Four hypothetical scenarios of trade flows after sanctions. Solid and dotted arrows indicate hypothetical premises of trade-augmenting and trade-diminishing results, respectively.

**Figure 2 foods-12-03934-f002:**
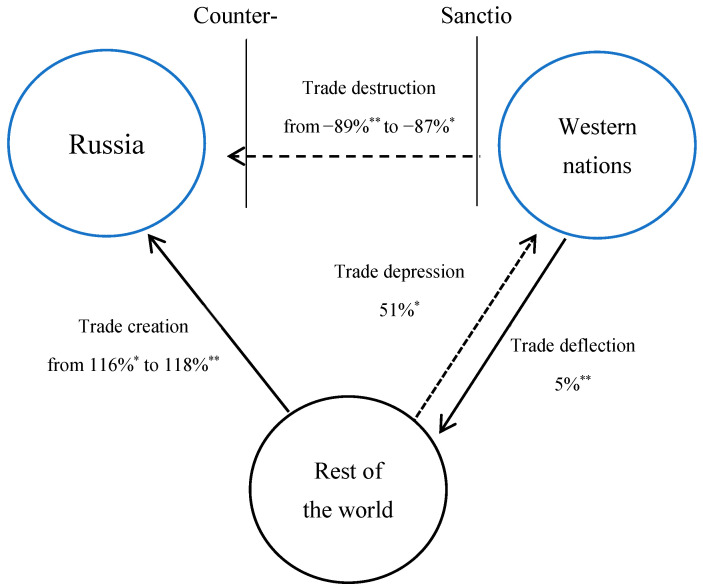
Trade impacts per trade pathway using the fixed effects. * and ** designate the estimates made using Model (c) and Model (d), respectively.

**Table 1 foods-12-03934-t001:** Main seafood exporters to Russia in the 2000s: Market share in percentage.

Country	2002	2005	2010	2011	2012	2013	2014	2015	2016	2017	2018	2019	2020
Norway	50.5	47.1	44.2	41.3	44.8	39.9	22.1	0.6	0	0.4	0.6	1	1.4
Chile	1.6	3.1	3.8	4.7	4.9	10.4	16	24.3	23.5	20.2	25.6	21.5	19.9
China	2.7	4	9.5	10.7	9.6	9.2	12	12.2	15.5	14.6	14.9	14.9	11
Vietnam	0.7	2.4	3.7	4.2	3.3	2.8	3.6	5.6	6.4	6.2	4.8	6	6.6
Argentina	0.2	3.2	1	1.3	0.9	0.8	1.4	1.5	2.2	3.4	3.5	4.7	4.9
India	0.2	0.1	1	1.3	1.2	1.4	2.9	2.3	4.2	4.2	4.9	5.4	5.1
Faroe Isl.	0.1	0.3	0.7	1.3	3.3	3.9	6.7	20.4	20.1	22.7	18.7	17.7	16.8
Turkey	0	0	0.4	1	0.9	1.2	2.4	3.2	2.8	3	4.1	5.2	7.9
Greenland	0.1	0.1	0.3	0	0	0	1	4.4	5.4	5.9	3.9	2.2	3.1
Belarus	0	0	0	0	0.8	2.2	4.3	7.1	7.4	7.2	6.5	6.6	6.5
Sum	56.1	60.3	64.6	65.8	69.7	71.8	72.4	81.6	87.5	87.8	87.5	85.2	83.2
Sum excluding Norway	5.6	13.2	20.4	24.5	24.9	31.9	50.3	81	87.5	87.4	86.9	84.2	81.8

Source: Trade statistics for international business development, https://www.trademap.org (accessed on 22 August 2020).

**Table 2 foods-12-03934-t002:** Selected studies on the consequences of sanctions on the economy.

Study	Sanctioned Effect	Data	Model
Caruso (2003) [18]	Averse bilateral trade between the US and 49 countries	Panel (1960–2020)	Gravity
Afesorgbor and Mahadevan (2016) [19]	Greater income inequality by 1.5–1.7 points in 68 states	Panel (1960–2008)	Regression with FE; GMM
Neuenkirch and Neumeier (2016) [20]	A 3.8%-point larger poverty gap in 85 states	Panel (1982–2011)	General equilibrium (GE)
Kholodilin and Netšunajev (2018) [22]	Small impact on GDPs and the exchange rates in Russia and the Eurozone	Panel (1997–2018)	Structural VAR
Dong and Li (2018) [21]	Optimal sanction and retaliation tariffs are calculated for the US, EU and Russia	Equilibrium dataset	16-country GE
Afesorgbor (2019) [23]	Adverse bilateral trade flows among 60 senders and 143 targets	Panel (1960–2009)	Gravity

Source: Authors’ compilation.

**Table 3 foods-12-03934-t003:** Selected studies on trade policy-related trade deflection.

Study	Trade Deflection	Data	Model
Bown and Crowley (2007) [24]	US anti-dumping led to a 5–7% increase in trade deflection for Japan.	Panel (1992–2001)	GMM, IV
Grant and Anders (2011) [14]	US import refusals for seafood generate a 1–13% increase in trade deflection toward rest-of-world markets.	Cross-section (1997, 2000, 2004 and 2006)	Gravity
Baylis et al. (2011) [25]	Each instance of EU import refusal of seafood reduces exports to the EU by 43% and generates trade deflection of a 3% import increase to non-EU states.	Panel (1998–2008)	Gravity
Cuello et al. (2020) [26]	Each EU import refusal of unauthorized GM foods generates trade deflection by 3.2%, 2.6%, and 1.8% for papaya, fructose, and cereals, respectively.	Panel (2008–2014)	Gravity

Source: Authors’ compilation.

**Table 4 foods-12-03934-t004:** Description of variables.

Variables	Description
$X_{ijt}$	Seafood (HS:03) importing value to importer i from exporter j at year t
${GDP}_{i\;or\;j,\;t}$	Gross domestic product of importer i or exporter j at time t (in current billion USD)
${DIST}_{ij}$	Distance between country i and j
${BOR}_{ij}$	Dummy variable takes the value ‘1′ if countries share contiguous borders, bilateral
${LNG}_{ij}$	Dummy variable takes the value ‘1′ if countries use common official language, bilateral
${COL}_{ij}$	Dummy variable takes the value ‘1′ if countries have common colonial past since 1945
${FTA}_{ij}$	Dummy variable takes the value ‘1′ if the pair currently has a free trade agreement
${T\_DES}_{ijt}$	Dummy variable takes the value ‘1′ if importer i is Russia and exporter j is WEST at year t from 2015 to 2020 (trade destruction)
${T\_DEF}_{ijt}$	Dummy variable takes the value ‘1′ if importer i is ROW and exporter j is WEST at year t from 2015 to 2020 (trade deflection)
${T\_CRE}_{ijt}$	Dummy variable takes the value ‘1′ if importer i is Russia and exporter j is ROW at year t from 2015 to 2020 (trade creation)
${T\_DEP}_{ijt}$	Dummy variable takes the value ‘1′ if importer i is WEST and exporter j is ROW at year t from 2015 (trade depression)

Note: The Harmonized System (HS) Code 03 pertains to ‘fish and crustaceans, mollusks and other aquatic invertebrates.’

**Table 5 foods-12-03934-t005:** Descriptive statistics of variables.

Variable	No. of Observation	Mean	Standard Deviation	Min	Max
Import values (USD)	544,640	2,751,547	4 × 10^7^	0	4.78 × 10^9^
GDP (USD)	544,640	3.89 × 10^11^	1.55 × 10^12^	2.18 × 10^7^	2.14 × 10^13^
Distance (Km)	544,640	8134.14	4576.79	10.48	19,951.16

**Table 6 foods-12-03934-t006:** Estimation results.

Variable	Model (a)	Model (b)	Model (c)	Model (d)
Ln (GDP-importer)	0.90094 *** (0.00969)	0.89898 *** (0.00946)	1.01245 *** (0.08461)	0.76354 *** (0.05149)
Ln (GDP-exporter)	0.55519 *** (0.00774)	0.55724 *** (0.00780)	0.01837 (0.05174)	0.12220 *** (0.03517)
Ln (distance)	−0.47714 *** (0.01810)	−0.48234 *** (0.01816)	−0.76109 *** (0.01667)	-
Contiguity	0.58152 *** (0.07605)	0.58153 *** (0.07641)	0.68826 *** (0.05249)	-
Common language	−0.16427 *** (0.05540)	−0.17047 *** (0.05526)	0.21173 *** (0.05096)	-
Colonial past	0.28740 *** (0.05043)	0.28552 *** (0.05067)	0.66520 *** (0.06082)	-
FTA	0.37875 *** (0.03630)	0.36600 *** (0.03671)	0.10234 *** (0.02642)	−0.25122 (0.62499)
Trade destruction		−2.0318 *** (0.58509)	−2.02120 *** (0.52259)	−2.24318 *** (0.59137)
Trade deflection		−0.34586 ** (0.15377)	0.10148 (0.10570)	0.05345 * (0.02908)
Trade creation		0.26000 ** (0.12505)	0.77603 *** (0.12909)	0.78520 *** (0.11869)
Trade depression		0.04327 (0.05238)	0.41699 *** (0.04703)	−0.02228 (0.02534)
Constant	−19.48821 *** (0.45369)	−19.43789 *** (0.44908)	−4.68920 ** (2.29727)	−5.6779 *** (1.60282)
Country–time FE	No	No	Yes	Yes
Country–pair FE	No	No	No	Yes
Pseudo R^2^	0.6029	0.6038	0.8568	0.9746
Observation	544,640	544,640	541,696	214,496

Note: (1) * *p* < 0.1, ** *p* < 0.05, *** *p* < 0.01. (2) Robust SE is in parenthesis.

## Data Availability

The data presented in this study are available on request from the corresponding author.

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
