# Peer review of "Assessing the Seafood Trade Diversion Arising from Economic Sanctions: Evidence from Russia and Western Countries"

_foods, 2023, doi:10.3390/foods12213934_

Round 1
Reviewer 1 Report
Comments and Suggestions for Authors
After reading the manuscript several times it is still very difficult to see how it binds together and thus also how it should contribute to the academic community. First - the title is not easy to understand - what is the focus of this manuscript? It probably miss some critical words for the sentence to make sense. The introduction is very brief and seems detached from the calculations based on a structural gravity model. Consequently the conclusion is also detached from the intro, the very slim literature review and the calculations.
Comments on the Quality of English LanguageThe concern about the quality of English language is related to the fact that many passages are difficult to understand. For example the sentence stating at line 14; According to estimates....
Reviewer 2 Report
Comments and Suggestions for Authors
Overall the paper reads very well. I have a few relatrively minor comments that should be considered when revisiing the paper:
Line 49: “Doctrine” rather than “Doctorate”
Table 1. it might be worth including another row below Sum, with “Sum excluding Norway”. This would highlight the shift as it would then go from 6% to 82% for the other countries listed.
Line 112: “Russia used to export …” should be “Russia used to import …”
Section 5 – it might be worth highlighting in the text that robust standard errors were estimated to counter problems of heteroskedasticity (was this actually observed in the model?). Also, fixed effects is a good assumption, but were random effects also considered and then formally tested to ensure that fixed effects was the most appropriate assumption? This would not take a lot of time to re-run the models as random effect and then apply a Hausman test to see which assumption is most appropriate.
